# CFD Modeling of Spatial Inhomogeneities in a Vegetable Oil Carbonation Reactor

**Attila Egedy [1,\*], Alex Kummer [1], Sébastien Leveneur [2] , Tamás Varga [1] and Tibor Chován [1]**

[1]   Department of Process Engineering, University of Pannonia, 10 Egyetem Str., H-8200 Veszprém, Hungary; kummera@fmt.uni-pannon.hu (A.K.); vargat@fmt.uni-pannon.hu (T.V.); chovan@fmt.uni-pannon.hu (T.C.)

[2]   Laboratoire de Sécurité des Procédés Chimiques LSPC, EA4704, INSA Rouen, Normandie University, UNIROUEN, 76800 Saint-Étienne-du-Rouvray, France; sebastien.leveneur@insa-rouen.fr

\*   Correspondence: egedya@fmt.uni-pannon.hu

**Abstract:** Fossil materials are widely used raw materials in polymerization processes; hence, in many cases, the primary goal of green and sustainable technologies is to replace them with renewables. An exciting and promising technology from this aspect is the isocyanate-free polyurethane production using vegetable oil as a raw material. Functional compounds can be formed by the epoxidation of vegetable oils in three reaction steps: epoxidation, carbonation, and aminolysis. In the case of vegetable oil carbonation, the material properties vary strongly, with the composition affecting the solubility of $CO_2$ in the reaction mixture. Many attempts have been made to model these interactions, but they generally do not account for the changes in the material properties in terms of spatial coordinates. A 2D CFD model based on the combination of the k-ε turbulence model and component mass balances considering the spatial inhomogeneities on the performance of the reactor was created. After the evaluation of the mesh independence study, the simulator was used to calculate the carbonation reaction in a transient analysis with spatial coordinate-dependent density and viscosity changes. The model parameters (height-dependent mass transfer parameters and boundary flux parameters) were identified based on one physical experiment, and a set of 15 experiments were used for model validation. With the validated model, the optimal operating temperature, pressure, and catalyst concentration was proposed.

**Keywords:** carbonation modeling; spatial coordinate-based material properties; biomass valorization; CFD

## 1. Introduction

Fossil raw materials are the most commonly used raw materials in the energy sector and the chemical industry. Nearly 90% of organic matter-based products are produced from petroleum or natural gas. The cornerstone of sustainable technology development is to reduce the dependence on fossil raw materials, which is an essential directive at the industrial and governmental levels as well. Using vegetable oil as a renewable raw material is an excellent example of this effort. The most important characteristics of vegetable oil are its accessibility and excellent biodegradability. Different feedstocks can be used for the process—for instance, soybean, rapeseed, sunflower seed, palm oil etc. [1]. The properties of the vegetable oil vary with the length of the chains and the number of double bonds. The application areas of vegetable oils are vast; they are used to produce different type of polyurethanes, resins, and even composites, which make vegetable oils promising compounds in the future of chemistry [2]. One of the most promising fields can be the production of secondary fuel for diesel vehicles [3,4].

In this study, we focus on the production of widely used isocyanate-free polyurethane (this polymer is the sixth most commonly used material). Several other studies dealt with the investigation of

the reaction system; also, several kinetic models were derived. However, only some of the studies include the investigation of the varying material properties during the reaction and component transfer processes, such as the viscosity and the density. In case of the carbonation at low temperatures, the viscosity can increase by 30–50%, while the density can increase by about 5%. Cai et al. considered the viscosity and density changes in their model [5]. However, spatial inhomogeneities were not considered. From the three reaction steps (epoxidation, carbonation, and aminolysis) of polyurethane production, we considered the carbonation reaction for a detailed investigation and optimization. The carbonation reaction often takes place at mild temperatures [6], where the temperature is between 110 and 150 °C, the residence time is about 8 h, and the pressure is between 30 and 60 bar.

A simple mathematical model can be used for parameter identification and optimization of the process. However, with a concentrated parameter model, it is impossible to account for the spatial inhomogeneities in a complex system. The more detailed Computational Fluid Dynamics (CFD) models can be applied for these problems with the tradeoff of a much higher computational time. CFD models are well suited to understand the underlying processes and compare different reactor constructions. However, in most cases, the term optimization in the CFD context often means the comparison of different scenarios (such as experimental optimization) [7], rather than the application of an objective function-based optimization. If an adequate model can be implemented in 3D or 2D with a reasonable amount of computational time, then ordinary optimization methods can also be applied.

In this study, a 2D CFD model was proposed to calculate the spatial distribution of the process variables during the carbonation reaction. The velocity field of the 2D representation was compared to a 3D simulation to make sure that the velocity field is similar and the 2D simulator is applicable for the task. The kinetic model is based on the work of Cai et al. [5] with additional terms to describe the $CO_2$ flux between the gas and the liquid phase. The height-dependent mass transfer coefficients, as well as the viscosity and the density, are based on the spatial coordinates. The model parameters were identified, and the model was validated against multiple measurements. The detailed model describes the experimental data more adequately, and it can predict the behavior of the system appropriately. In the following, we determined the optimal temperature, pressure, and initial catalyst concentration to maximize the conversion of the reaction.

## 2. Reactor Model Development

This section presents the geometry selection (see Section 2.1), the reaction modeling (see Section 2.2), and the CFD model of the system, including the mesh independence study (see Section 2.3). We also introduce the experiments for the validation of the model (see Section 2.4).

### 2.1. Geometry Selection, Momentum Balance Calculation for the 2D and the 3D Model

The reactor was a Parr 300 mL reactor with 100 mm inner diameter equipped with a gas entrainment impeller (25 mm diameter). At the beginning of each experiment, 100 mL of epoxidized cottonseed oil (ECSO) was loaded into the reactor. Further information about the measurements can be found in Cai et al. [5]. A 500 rpm revolution speed was applied in each experiment. A 3D and a 2D representation of the reactor geometry were implemented in the CFD model. In the case of the 3D representation, a rotating reference frame method was used, while swirl flow was implemented in the case of the 2D representation. In this study, an axisymmetric 2D representation of the reactor was implemented in COMSOL Multiphysics. The main idea of the axisymmetric calculation is the facilitation of the symmetry boundaries; thus, we can decrease the computational time significantly. In this case, the reactor itself is not symmetric due to the asymmetry of the impeller. However, the relatively fast revolution speed (500 rpm) makes the flow field relatively symmetric, which justifies using this approach.

Figure 1a shows the 3D and the 2D representation, and Figure 1b shows the resulting velocity fields (m/s). The velocity fields were similar in the case of 3D and the 2D simulations. However, a multiplication factor of 2.8 has to be implemented in the revolution speed in case of

the 2D simulation. To make sure that the numerical values are similar in the case of the axial symmetry model, a reasonable agreement was found between the velocity field of the 3D and the 2D representation, so the 2D model was used for the further parameter identification and optimization steps. The main advantage of using the 2D model is the lower computation time, making the ordinary optimization methods applicable.

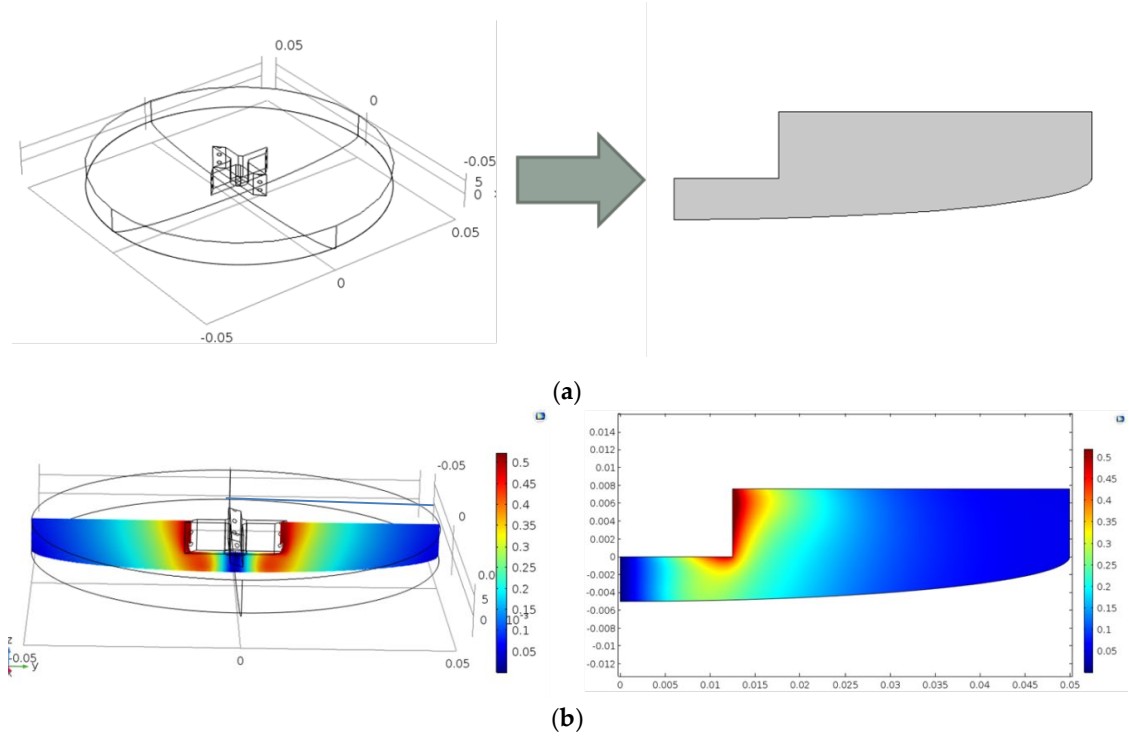

(**a**)

(**b**)

**Figure 1.** (**a**) The 3D and 2D representation of the reactor; (**b**) the velocity field results in the case of 3D and 2D simulation.

## 2.2. Carbonation Reaction Modeling

Figure 2a shows the reactants (epoxidized cottonseed oil, ECSO) and the products (carbonated cottonseed oil, CCSO). Figure 2b presents the reaction mechanism. In this work, the model of Cai et al. was extended to consider spatial inhomogeneities. Later on, we will refer to this model as the original one [5].

The viscosity of the reaction mixture, which can be calculated based on the sum of weighted viscosities of the components varies in the spatial and temporal coordinates. Equation (1) presents how the reaction rate is calculated.

$$R_{carbonation} = \frac{k_{carbonation} \cdot \left[CO_{2,liq}\right] \cdot [TBAB]^n ([ECSO] + \gamma [CCSO])}{\alpha + \beta \cdot \left[CO_{2,liq}\right]} \tag{1}$$

(**a**)

(**b**)

**Figure 2.** The applied reaction mechanism (**a**) simplified (**b**) detailed with catalyst [5].

The maximum solubility of $CO_2$ ($c_{CO2\_liq*}$) is calculated within the CFD simulator based on the Henry coefficients introduced in the original model. Equations (2) and (3) show the calculation of $c_{CO2\_liq*}$ and the overall He coefficient.

$$c_{CO2\_liq*} = He_i \cdot p_{reac} \tag{2}$$

$$He_{mixing} = x_{ECSO} \cdot He_{ECSO} + x_{CCSO} \cdot He_{CCSO} \tag{3}$$

Equation (4) shows the first addition to the original model, the flux of the $CO_2$ at the inlet boundary, which was the contact area between the fluid and the gas phase.

$$D_{flux} = D\left(c_{CO2\_liq*} - c_{CO2\_liq}\right) \tag{4}$$

The component viscosities and densities are calculated with the following Equations (5)–(8), based on the measurements described in Cai et al. [5].

$$\mu_{ECSO} = 684.41 \cdot \exp(-0.028 \cdot T_{reac}) \tag{5}$$

$$\mu_{CCSO} = 3 \cdot 10^7 \exp(-0.055 \cdot T_{reac}) \tag{6}$$

$$\rho_{CCSO} = -0.6958 \cdot T_{reac} + 1236.4 \tag{7}$$

$$\rho_{ECSO} = -0.6904 \cdot T_{reac} + 1183.1 \tag{8}$$

The main modification regarding the $\mu_{mix}$ and $\rho_{mix}$ term was that the calculation was not only based on time but also on spatial coordinates, which were calculated in all of the node points.

$$\mu_{mix} = \exp(w_{ECSO} \cdot \log(\mu_{ECSO}) + w_{CCSO} \cdot \log(\mu_{CCSO})) \tag{9}$$

$$\rho_{mix} = w_{ECSO} \cdot \rho_{ECSO} + w_{CCSO} \cdot \rho_{CCSO}. \tag{10}$$

The material properties were implemented using Equations (9) and (10), including temperature and coordinate-based calculation. Another extension of the original model is the calculation of the $N_{CO2}$ term, where the mass transfer coefficient and the vertical change in the bubble size were considered, where $z$ refers to the spatial $z$ coordinate. $A$ and $B$ are the constants to be identified.

$$N_{CO2} = (A \cdot kLA + B \cdot (z + 0.005)) \cdot \left(c_{CO2\_liq*} - c_{CO2\_liq}\right) \tag{11}$$

Equations (12)–(14) present the component balances, which are in similar forms as in the original model.

$$\frac{dC_{Ep}}{dt} = -R_{carbonation} \tag{12}$$

$$\frac{dC_{Carb}}{dt} = R_{carbonation} \tag{13}$$

$$\frac{dC_{CO2\_liq}}{dt} = -R_{carbonation} + N_{CO2} \tag{14}$$

### 2.3. CFD Model of the Reactor

One of the main goals was to calculate the spatial and temporal changes of the viscosity and the density. In most of the CFD simulators, when the momentum balance is close to stationary (or reaches the stationary state significantly faster than the component balance) the two-equation sets can be separated, and a stationary momentum balance can be used as a basis for the calculation of the component balances. In our case, the viscosity and the density, which are the two main material properties used in the momentum balance, change in time. Since they vary with the progression of the reaction, the two balances cannot be separated. Thus, a longer computational time is expected.

The reactor was operated isothermally, so we neglected the solution of the heat balances. A mesh independence study was performed to ensure the validity of the model. Five different mesh sizes were applied, and the changes in the velocity field were calculated within a grid. The changes in the velocity field are getting lower by the increase of the number of mesh elements, and consequently, it results in a higher computational time. After the 4th mesh size, the changes in the velocity become around 1%, while the computational time tripled from 4th to 5th mesh, so the 4th mesh size was chosen for the further calculations. Figure 3 shows the results of the mesh independence study and the resulting mesh.

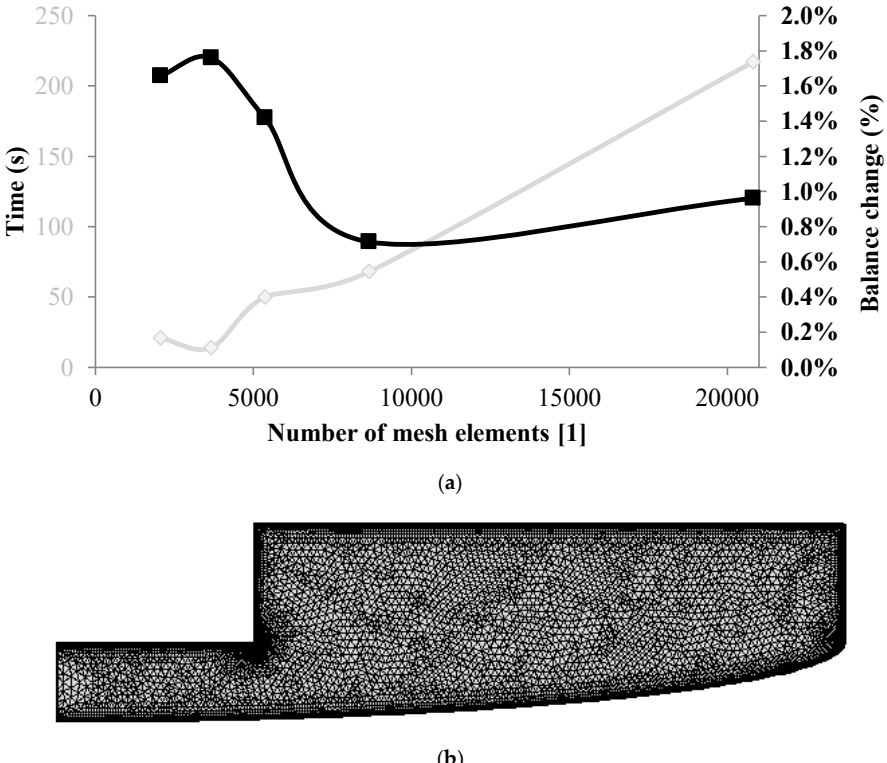

(**a**)

(**b**)

**Figure 3.** (**a**) the Results of the mesh independence study; (**b**) the applied mesh (8655 elements).

### 2.4. Experimental Work Considered

Based on a previous article [5], 14 experiments were considered for this study. The investigated temperature interval was between 110 and 140 °C; the pressure was between 30.6 and 52 bar. Table 1 presents the operating conditions of the experiments.

**Table 1.** The experiments considered for this study (only the cases where epoxidized cottonseed oil (ECSO) was present initially were considered).

| Number of Experiment | Initial ECSO Concentration (mol/L) | Initial Catalyst Concentration (mol/L) | Pressure (bar) | Temperature (°C) |
|---|---|---|---|---|
| 1 | 3.6 | 0.13 | 30.6 | 120 |
| 2 | 3.27 | 0.13 | 32 | 110 |
| 3 | 3.38 | 0.3 | 21.1 | 140 |
| 4 | 3.36 | 0.13 | 48.2 | 140 |
| 5 | 3.26 | 0.06 | 48.8 | 130 |
| 6 | 3.41 | 0.13 | 45.8 | 140 |
| 7 | 3.53 | 0.13 | 30.3 | 120 |
| 8 | 3.34 | 0.13 | 45.9 | 140 |
| 9 | 3.8 | 0.13 | 40.4 | 120 |
| 10 | 3.44 | 0.13 | 49.7 | 120 |
| 11 | 3.22 | 0.2 | 48.1 | 140 |
| 12 | 3.4 | 0.13 | 46.8 | 110 |
| 13 | 3.06 | 0.3 | 48.5 | 140 |
| 14 | 3.29 | 0.13 | 52 | 120 |

## 3. Results

This section introduces the results of the model parameter identification (see Section 3.1) and the results of the optimization problem (see Section 3.3). Section 3.2 describes the sensitivity analysis of the revolution speed based on the first experiment.

### 3.1. Model Parameter Identification

Three parameters were identified, which are the A and B height-dependent mass transport parameters in Equation (12), and the D component transfer parameter from Equation (4) using experiment 1 (see Table 1). The NOMAD black-box optimization toolbox was used for parameter identification [8,9]. The following objective function (Equation (15)) is applied to identify the values of the A, B, and D parameters, which is a sum of the relative squared errors in time.

$$Error = \sum_{k=1}^{m} \left( \left( c_{ECSO\_experimental} - c_{ECSO\_simulation} \right) / c_{ECSO\_simulation} \right)^2 \tag{15}$$

where $k$ is the number of measurement points (in time).

The upper and lower limits for the parameters A, B, and D were $(3, 15, 3 \times 10^{-4})$ and $(1, 10, 10^{-4})$, respectively. A COMSOL-MATLAB Livelink connection was used for the calculation, where the 2D CFD model was calculated in every function call. An Intel Xeon E5620 computer was used for the calculation with 72 GB RAM. One function call lasted around five minutes. A grid size-based integrated concentration was used to compute the simulated value. The parameter identification step resulted in A = 2.101, B = 13.223, and D = $1.984 \times 10^{-4}$ (m/s). The summed error was 9.45.

Figure 4 shows the integrated ECSO concentrations. The markers show the measured values for each experiment, while the continuous lines show the simulated values.

As can be seen in Figure 4, most of the simulated values are in good agreement with the measured values. To further evaluate the effect of the process parameters of the errors, the connection between the temperatures, pressures, and process parameters are plotted in Figure 5.

As we can see, the model error is higher at lower temperatures (see Figure 4; exp. 2 and 12). The higher pressures correlate with higher errors (see Figure 4; exp. 5, 11, 12 and 13).

In the next section, the effect of the temperatures and pressures are shown on the velocity field and the concentrations profiles. Firstly, we show the results of experiments 3 and 13, which have the same temperature, and then experiments 8 and 12, which have similar pressure.

Figure 6 presents the velocity field and the concentration of ECSO and $CO_2$ in the case of experiments 3 and 13. The temperature is the same in both experiments, but the pressure of experiment 13 is twice as much as in experiment 3. The velocity fields and ECSO and $CO_2$ concentrations are shown at the end of the simulation time.

The velocity field is different only in the maximum velocity; the higher pressure will lead to higher maximum velocity (0.721 compared to 0.717) as well as higher $CO_2$ concentration. The concentrations fields are relatively homogeneous.

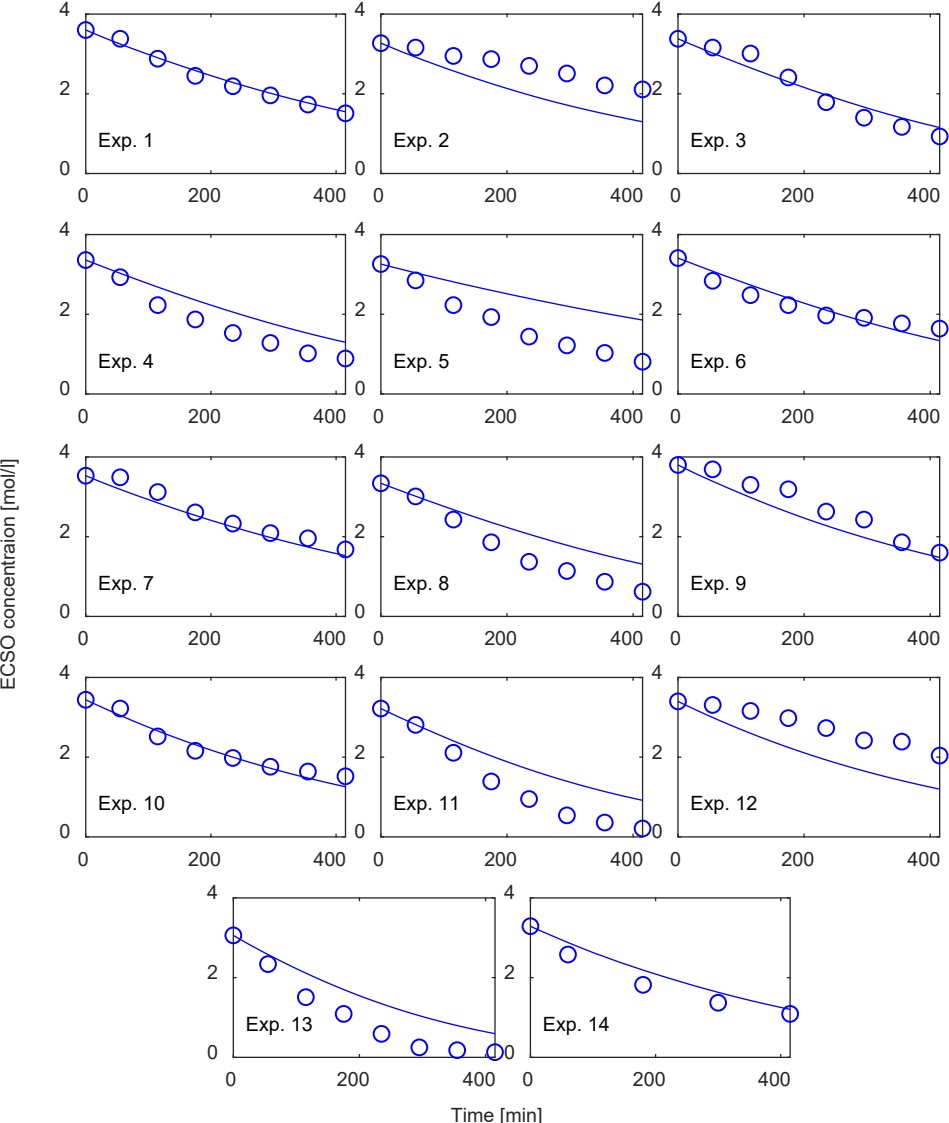

**Figure 4.** The comparison of the measured (marker) and the simulated results.

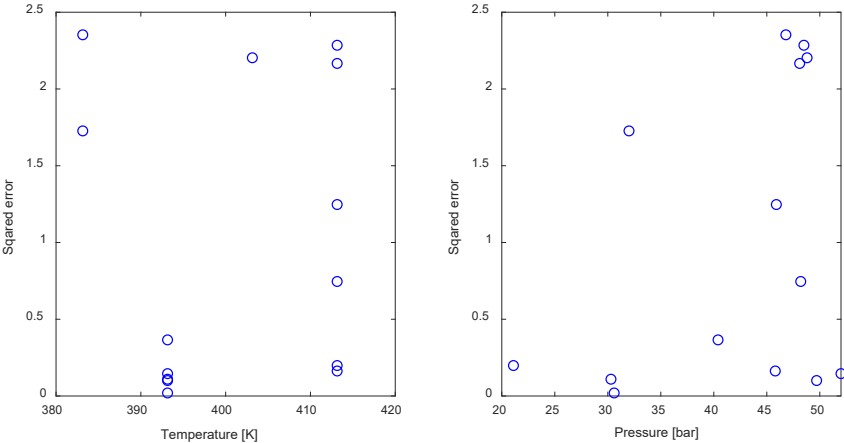

**Figure 5.** The squared errors plotted against the temperature and the pressure.

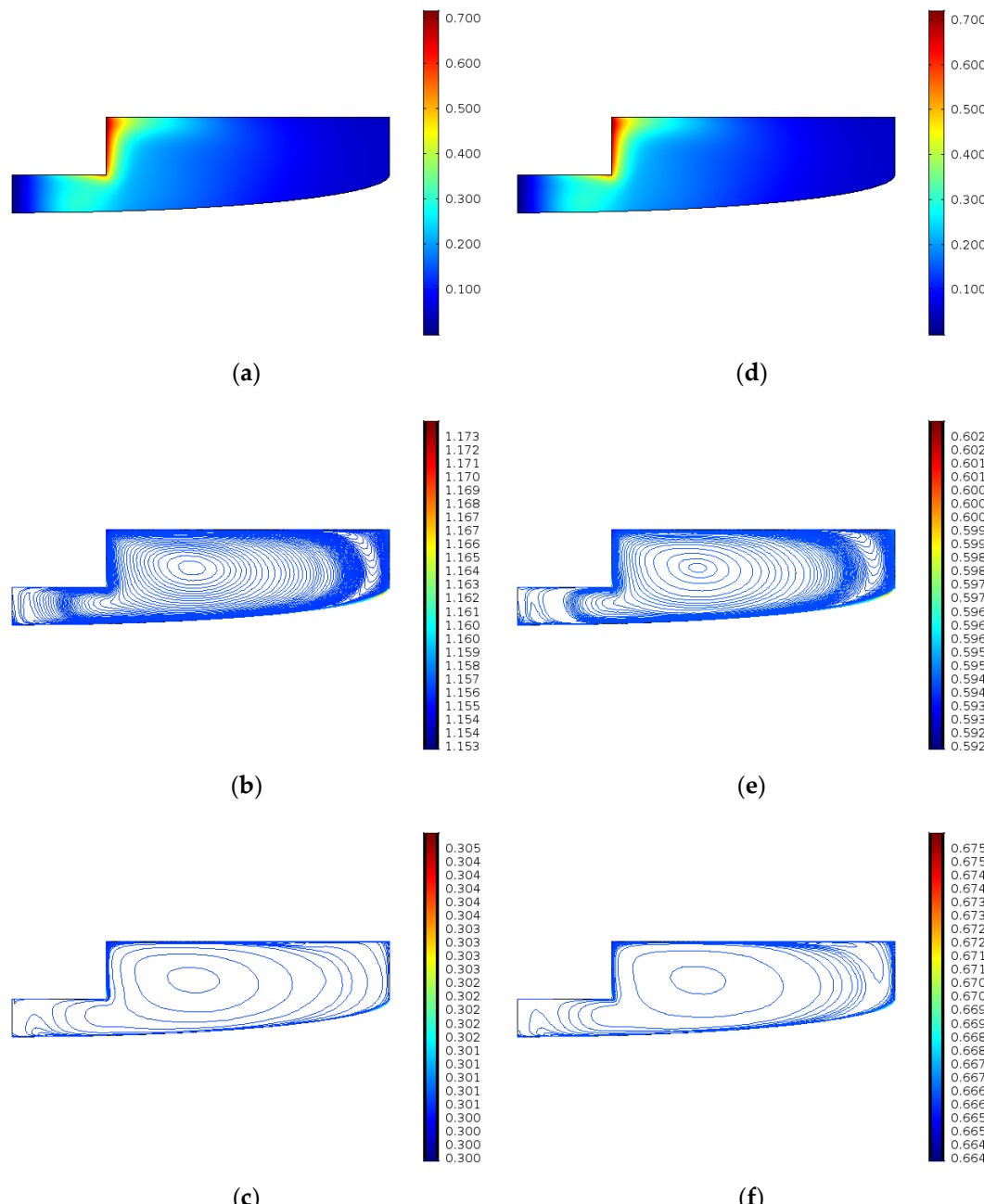

**Figure 6.** Results after 4 h (**a**) velocity field (m/s) in case of experiment 3; (**b**) ECSO concentration (mol/m$^3$) in case of experiment 3; (**c**) CO$_2$ concentration (mol/m$^3$) in case of experiment 3; (**d**) velocity field (m/s) in case of experiment 13; (**e**) ECSO concentration (mol/m$^3$) in case of experiment 13; (**f**) CO$_2$ concentration (mol/m$^3$) in case of experiment 13.

Figure 7 shows the velocity field and the concentrations of ECSO and CO$_2$ in the case of experiments 8 and 12. The pressure is similar, but the temperature of experiment 12 is 383.15 K, while in the case of experiment 8, it is 413.15 K. The results are shown at the end of the simulation time. As we can see, the velocity field of experiments 8 and 12 are different, meaning that the temperature has a more significant effect on the velocity field than pressure.

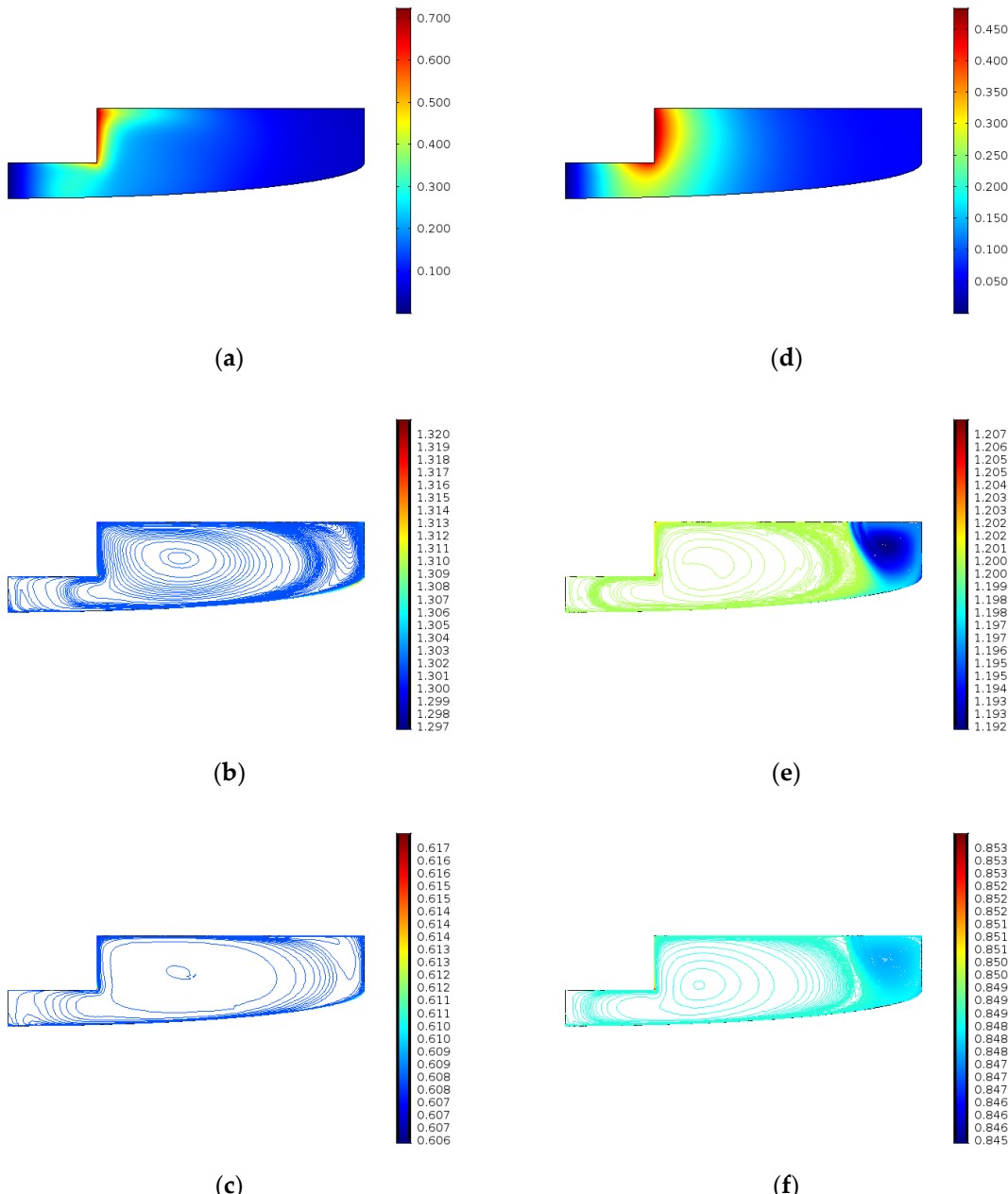

**Figure 7.** Results after 4 h (**a**) velocity field (m/s) in case of experiment 8; (**b**) ECSO concentration (mol/m$^3$) in case of experiment 8; (**c**) $CO_2$ concentration (mol/m$^3$) in case of experiment 8; (**d**) velocity field (m/s) in case of experiment 12; (**e**) ECSO concentration (mol/m$^3$) in case of experiment 12; (**f**) $CO_2$ concentration (mol/m$^3$) in case of experiment 12.

The ECSO and the $CO_2$ concentration profiles get more homogenous with the increase of temperature compared to the lower temperature cases.

The viscosity and the densities are time and spatial coordinate dependent. Figure 8 shows the viscosity and density changes in the function of time in the case of experiment 1. As we can see, the dynamic viscosity of the reaction phase changes by around 7%, while the density change is around 3%. The changes in time are significant, but the changes in space are minimal due to adequate mixing.

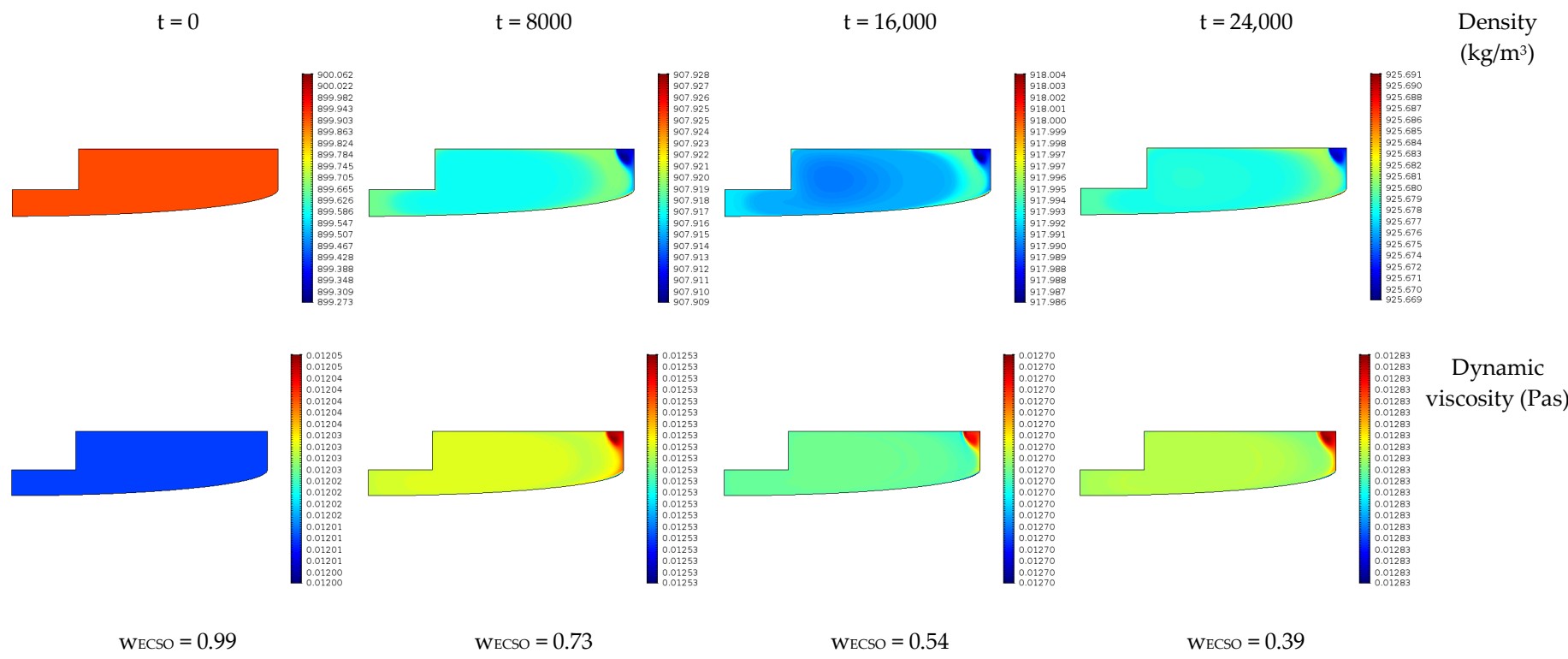

**Figure 8.** The changes of viscosity and density in time for experiment 1 (and consequently ECSO concentration).

### 3.2. Sensitivity Analysis of the Revolution Speed (Experiment 1)

In this section, we examined the effect of the different revolution speeds. The calculation was completed with the simulator, including the component mass balances.

In addition to the initial case, two different revolution speeds were applied (100 rpm and 300 rpm). Figure 9a–c shows the velocity fields.

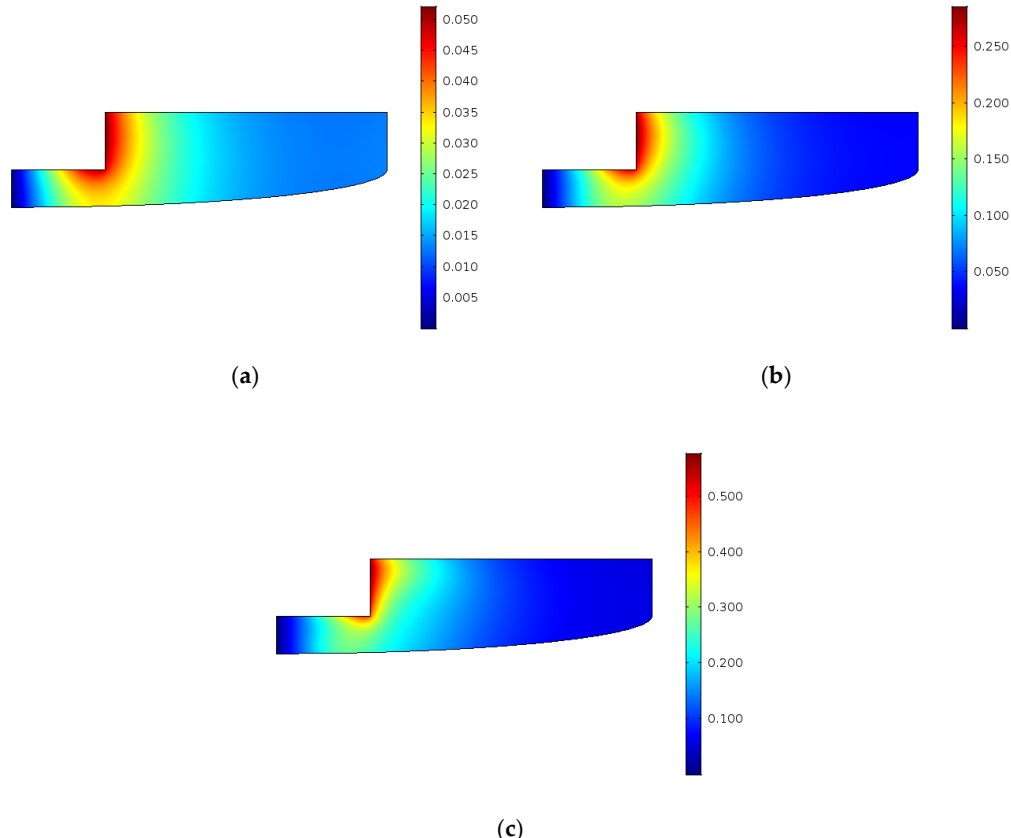

(**a**)                                             (**b**)

(**c**)

**Figure 9.** The sensitivity analysis of the revolution speed (**a**) 100 rpm; (**b**) 300 rpm; (**c**) 500 rpm.

As we can see, not only the magnitude but the characteristics of the flow are changing with the increase of the revolution speed. The changes of ECSO concentration were minimal. The change of the velocity magnitude is not linear dependent on the revolution speed; however, to extend the model for operating with different revolution speeds, additional measurements are needed.

### 3.3. Optimization

The reactor performance was optimized, considering the three most critical process parameters: the process temperature, the pressure, and the initial catalyst concentration. The NOMAD black-box optimization algorithm was applied to solve the optimization problem. COMSOL-MATLAB Livelink was used for the solution. The CFD simulator was implemented in COMSOL, while the optimization algorithm was implemented in a MATLAB environment. The upper and the lower limits of the optimization problem were (413.15 60 0.7) and (383.15 30 0.5), respectively. The temperature and pressure boundaries were defined based on the physical limitations of the systems. The objective is to maximize the concentration of the carbonated cottonseed oil (CCSO) at the end of the reaction. The optimal parameters were found as 383.39 K and 51.14 bar, while the optimal initial concentration of the catalyst is 0.699 mol/L. The outlet concentration of ECSO is $1.36 \times 10^{-5}$ mol/L. Figure 10 shows the integrated concentrations of epoxidized cottonseed oil (ECSO) and the carbonated cottonseed oil (CCSO).

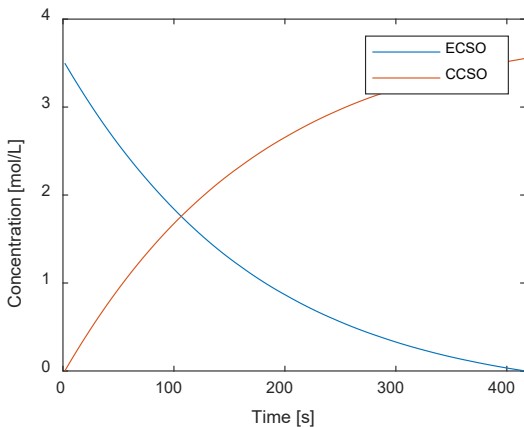

**Figure 10.** Integrated concentrations of ECSO and CCSO with the optimal operating parameters 383.39 K, 51.14 bar, 0.699 mol/L.

As we can see, full conversion can be achieved with the optimal parameters. Figure 11 shows the velocity field and the concentration of ECSO within the reactor after 4 h.

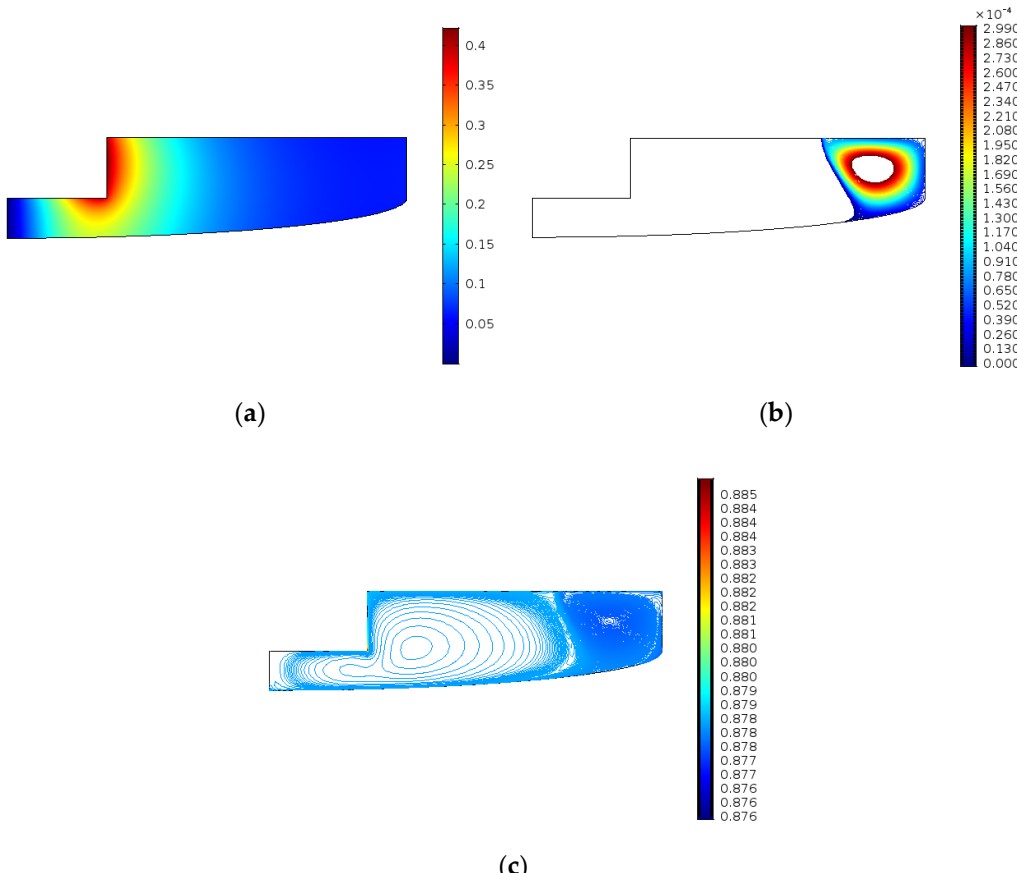

**Figure 11.** Results in case of the optimal solution after 4 h: (**a**) velocity field (m/s); (**b**) ECSO concentration (mol/m$^3$); (**c**) $CO_2$ concentration (mol/m$^3$).

The velocity field is similar in low-temperature cases. The ECSO concentration is reduced to zero almost in the whole reactor volume, only a minimal amount remains. However, the main driving force for the higher conversion is the higher concentration of $CO_2$ in the mixture, which is the result of the higher pressure. Figure 12 shows the changes in viscosity and density in time. As the reaction progresses further than the experimental cases, the changes of viscosity and density are higher as well.

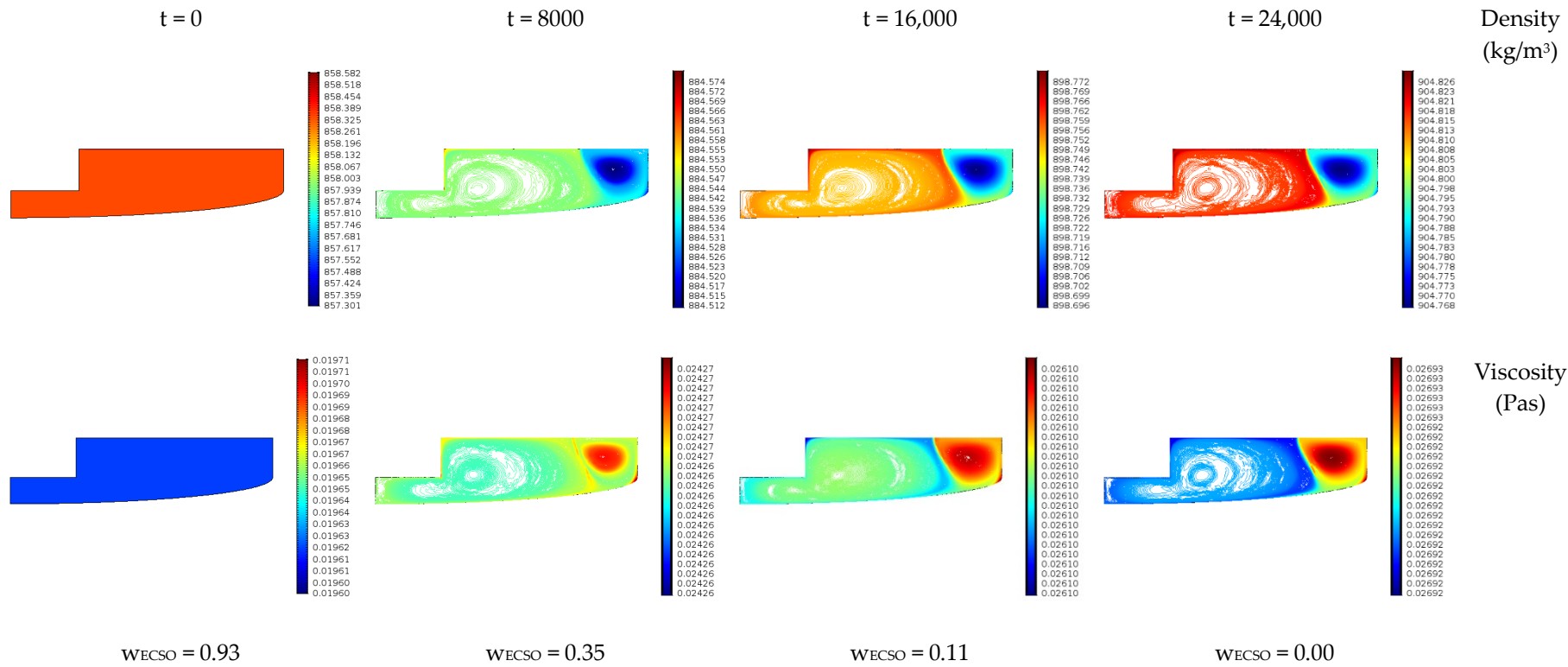

**Figure 12.** The changes of viscosity and density in time for the optimal solution (and consequently ECSO concentration).

Figure 12 shows the spatial coordinate-dependent changes in density and viscosity in case of the optimum. The viscosity increase is 35%, while the density increase is around 5%. As we can see, the viscosity depends on the ratio of ECSO/CCSO. In the case of lower temperatures and high pressure, the viscosity changes affect mainly the velocity field. The developed methods can serve well even with lower revolution speeds; however, to validate low revolution speed results, additional measurements will be needed.

## 4. Conclusions

In this study, a 2D CFD model of a carbonation reactor was developed. The model was based on an axial symmetric approximation of the system using the swirl flow model after the comparison of the 3D counterpart. The turbulent momentum balance was calculated together with the component mass balances in a transient simulation calculating the spatial coordinate-dependent changes of viscosity and density in the function of the component concentration, which is the main contribution of this study to the field.

The model parameters were identified based on one experiment and validated against 13 distinct measurements minimizing a relative squared error between the experimental and simulation results. The changes in concentrations, velocity fields, and the material parameters are discussed.

Then, the validated simulator is applied to optimize the optimal operating parameters, which are 383.39 K, 51.14 bar, and 0.699 mol/L concentration of the catalyst.

**Author Contributions:** Conceptualization, A.E. and T.V.; methodology, S.L. and A.E.; software, A.E., T.V.; validation, S.L., A.E. and A.K.; writing—original draft preparation, A.E.; writing—review and editing, T.C., T.V., A.K. and S.L.; visualization, A.E. and T.V.; project administration, A.E.; funding acquisition, A.E. and T.C. All authors have read and agreed to the published version of the manuscript.

**Funding:** Project no. 2019-2.1.11-TÉT-2019-00005 has been implemented with the support provided from the National Research, Development and Innovation Fund of Hungary, financed under the 2019-2.1.11-TÉT funding scheme. Tamás Varga's contribution to this paper was supported by the Janos Bolyai Research Scholarship of the Hungarian Academy of Sciences and UNKP-20-5 New National Excellence Program of the Ministry for Innovation and Technology from the source of the National Research, Development and Innovation Fund. The authors also thank the PHC program Balaton "Numerical methods for the optimization and safe production of non-isocyanate polymers" (44587VF).

**Conflicts of Interest:** The authors declare no conflict of interest.

## Abbreviations

| | |
|---|---|
| w | weight percent |
| x | molar percent |
| $\alpha$ | $k_1 \cdot k_2 \cdot k_3$ ($s^{-2}$) |
| $\beta$ | $k_2 \cdot k_3$ (L$\cdot$ mol$^{-1}$.s$^{-2}$) |
| $\gamma$ | $k_{-2}/k_1$ (mol$\cdot$L$^{-1}$) |
| $\mu$ | liquid viscosity (Pa$\cdot$s) |
| $\rho$ | mass density (kg/m$^3$) |
| He | Henry's coefficient (mol$\cdot$L$^{-1}\cdot$bar$^{-1}$) |
| c | Concentration (mol$\cdot$m$^{-3}$) |
| CCSO | carbonated cottonseed oil |
| ECSO | epoxidized cottonseed oil |
| TBAB | tetra-n-butylammonium bromide |
| A, B | height dependent mass transfer constants |
| D | flux parameter |
| $k_{carbonation}$ | carbonation rate of reaction ($s^{-1}$) |
| $R_{carbonation}$ | carbonation source term (mol$\cdot$m$^{-3}\cdot$s) |
| $N_{CO2}$ | $CO_2$ solution term (mol$\cdot$m$^{-3}\cdot$s) |
| $T_{reac}$ | Reaction temperature (K) |

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
