# Peer review of "CFD Modeling of Spatial Inhomogeneities in a Vegetable Oil Carbonation Reactor"

_processes, doi:10.3390/pr8111356_

Round 1

Reviewer 1 Report

Review of the manuscript ID number : processes-969409

Entitled : CFD modelling of spatial inhomogeneities in 2 carbonation reactor

General comments:

  • In general, the paper topic is in line with the scope of “processes” journal.
  • The scope and the type of cited literature is appropriate.

Mainor revisions are needed to improve the quality of this manuscript before acceptance .

  • Please compare your results with similar CFD investigation results in carbonation reactors.
  • Please mention DOIs in the references where it is possible

Author Response

In general, the paper topic is in line with the scope of “processes” journal.

The scope and the type of cited literature is appropriate.

Minor revisions are needed to improve the quality of this manuscript before acceptance.

We would like to thank you for your work and suggestions. We tried to do our best to improve our paper based on your suggestions. In the following, we would like to present our answers to the raised issues as best as possible.

Q1 Please compare your results with similar CFD investigation results in carbonation reactors.

A1 We have performed a deep investigation of the literature, and we have not found any CFD related articles about carbonation of epoxidized vegetable oil. The only mention of CFD related to this field is found in the thesis work of Xiaoshuang Cai. (Xiaoshuang Cai Production of carbonated vegetable oils from a kinetic modeling to a structure reactivity approach. Chemical and Process Engineering. Normandie Université, 2019. English. ffNNT : 2019NORMIR05ff. fftel-02867962f).

In the conclusion section, the author mentions, that the CFD simulation can be an excellent next step in this research.

Nevertheless, there are articles about carbonation reactors, even with CFD models describing the carbonation of Calcium-oxide to Calcium-carbonate in a Ca-looping process. However, besides the similarity of the name, there is no connection with our research, since our aim is to describe the spatial inhomogeneities in the carbonation of cottonseed oil.

To make the reaction more clear we propose a new title as CFD modelling of spatial inhomogeneities in a vegetable oil carbonation reactor.

Q2 Please mention DOIs in the references where it is possible

A2 We checked the references and added the DOIs where it was available.

Besides the suggestions, we have found a typo in the omptimized temperature, 383.49 was corrected to 383.39.

A small error was found in case of Figure 4 and 5, the plots were redrawn, and the calculated error 14.012 was corrected to 9.45 (this is the relative squared error, which are shown now in Eq 15).

Reviewer 2 Report

This paper numerically investigated the carbonation reactor by using CFD. The results are useful for further understanding of the carbonation process. However, a few comments need to be addressed:

-The model uses some temperature and composition dependence of the density, diffusivity and viscosity. How about the thermal conductivity?

-Eq. 15: should it be (simulation-experiment) for the error calculation?

-The authors should show how this fitted rate expression can be used for other conditions apart from the data used to fit the model.

Author Response

This paper numerically investigated the carbonation reactor by using CFD. The results are useful for further understanding of the carbonation process. However, a few comments need to be addressed:

We would like to thank you for the reviewer’s work. We tried to do our best to improve our paper based on your suggestions. In the following, we would like to present our answers to the raised issues as best as possible.

Q1 The model uses some temperature and composition dependence of the density, diffusivity and viscosity. How about the thermal conductivity?

A1 We implemented an isothermal model since the experiments were close to isothermal (the temperature control assumed to be perfect; thus uniform temperature is assumed). The model equations contain diffusivity, density and viscosity but not thermal conductivity. The next step of our research will be the implementation of the heat balance. For this aim adiabatic measurements are necessary so that we can consider the heat of reaction, heat losses, and the heat capacity and thermal conductivity dependence.

Q2 Eq. 15: should it be (simulation-experiment) for the error calculation?

A2 Thank you for finding this mistake. We corrected it as follows:

Q3 The authors should show how this fitted rate expression can be used for other conditions apart from the data used to fit the model.

A3 The rate constants we used were fitted in earlier work (Reference 5 Cai, X., Zheng, J. L., Wärnå, J., Salmi, T., Taouk, B., & Leveneur, S. (2017). Influence of gas-liquid mass transfer on kinetic modeling: Carbonation of epoxidized vegetable oils. Chemical Engineering Journal, 313, 1168–1183. https://doi.org/10.1016/j.cej.2016.11.012). We describe this in detail in Section 2.2 (Carbonation reaction modeling).

The rate constant fitting and the results are accepted as adequate. In this study, we fitted A and B height dependent mass transport parameters in Eq. 12, and the D component transfer parameter in Eq. 4. Experiment 1 was used for the parameter identification, and all the other 13 experiments were used for the validation. In this aspect, we think that Figure 4 proves that our model works well beyond the parameter set of experiment 1, since the goodness of the fit is clearly visible.

Also, at the higher reaction temperature, TBABr can be decomposed in HBr, leading to the side reaction of ring-opening. This aspect was discussed in the article J.-L. Zheng, F. Burel, T. Salmi, B. Taouk, S. Leveneur, Carbonation of vegetable oils: influence of mass transfer on reaction kinetics, Industrial & Engineering Chemistry Research, 54(43) (2015) 10935-10944.

A 1.6 percent overall improvement was achieved with the proposed model extensions.

Besides the suggestions, we have found a typo in the optimized temperature, 383.49 was corrected to 383.39. A small error was found in case of Figure 4 and 5, the plots were redrawn, and the calculated error 14.012 was corrected to 9.45 (this is the relative squared error, which are shown now in Eq 15).

Round 2

Reviewer 2 Report

The previous comments have been addressed.